# Influence of Diabetes Mellitus and Nutritional Parameters on Clinical and Functional Aspects and Quality of Life in Patients Hospitalized Due to Exacerbation of Chronic Obstructive Pulmonary Disease

**DOI:** 10.3390/jcm12216874

**Published:** 2023-10-31

**Authors:** Cristhian Alonso Correa-Gutiérrez, Zichen Ji, Patricia Aragón-Espinosa, Sarah Rodrigues-Oliveira, Luyi Zeng, Olalla Meizoso-Pita, Cristina Sevillano-Collantes, Julio Hernández-Vázquez, Luis Puente-Maestu, Javier de Miguel-Díez

**Affiliations:** 1Respiratory Department, Gregorio Marañón General University Hospital, 28007 Madrid, Spain; cristcorrea9312@gmail.com (C.A.C.-G.); luis.puente@salud.madrid.org (L.P.-M.); jdemigueldiez@telefonica.net (J.d.M.-D.); 2Faculty of Medicine, Complutense University of Madrid, 28040 Madrid, Spain; patara01@ucm.es (P.A.-E.); sarahr07@ucm.es (S.R.-O.); 3Gregorio Marañón Biomedical Research Institute, 28007 Madrid, Spain; 4Endocrinology and Nutrition Department, Infanta Leonor University Hospital, 28031 Madrid, Spain; luyizeng33@gmail.com (L.Z.); olalla.meizoso@salud.madrid.org (O.M.-P.); cristina.sevillano@salud.madrid.org (C.S.-C.); 5Respiratory Department, Infanta Leonor University Hospital, 28031 Madrid, Spain; juliohernandezvazquez@hotmail.com

**Keywords:** chronic obstructive pulmonary disease, diabetes mellitus, nutrition, quality of life

## Abstract

Patients with chronic obstructive pulmonary disease (COPD) may experience exacerbations. During severe exacerbations, nutritional and endocrinological comorbidities can play an important role in the clinical and functional aspects of these patients. The aim of this study was to analyse the influence of the presence of diabetes mellitus (DM) and nutritional parameters on the deterioration of symptoms and quality of life during a severe exacerbation in patients with COPD. An observational study was conducted on COPD patients admitted due to an exacerbation. The COPD Assessment Test (CAT) questionnaire was administered, and clinical and functional parameters were compared based on the presence of nutritional and endocrinological alterations. A total of 50 patients were included, of whom 30 (60%) were male. The mean age was 70.5 years (standard deviation (SD) 9.6). The median CAT score during exacerbation was 25 (interquartile range (IQR) 17.5–30), and the baseline score was 13.5 (IQR 7–19), which represented a statistically significant difference (*p* < 0.001). Patients with iron deficiencies had a lower total CAT score (*p* = 0.041), specifically for items related to daily activity (*p* = 0.009) and energy (*p* = 0.007). Diabetic patients exhibited a greater decline in pulmonary function during exacerbation (*p* = 0.016), while patients with high thyroid-stimulating hormone (TSH) levels had a shorter hospital stay (*p* = 0.016). For COPD patients admitted due to an exacerbation, the metabolic assessment is useful and relevant in the clinical set-up, as endocrinological comorbidities negatively affect clinical and functional aspects of these patients.

## 1. Introduction

Chronic obstructive pulmonary disease (COPD) is characterized by a chronic and not fully reversible limitation of airflow due to alterations in the airways, which is accompanied by persistent respiratory symptoms caused by significant exposure to harmful gases or particles [1]. An exacerbation of COPD leads to a worsening of respiratory symptoms, including cough, sputum production, and dyspnea, necessitating pharmacological treatment, which exert a detrimental effect on patients’ quality of life and functional capacity for daily activities [1,2,3].

The COPD Assessment Test (CAT) questionnaire is a widely employed instrument for assessing various health-related domains and the well-being of COPD patients [4]. This questionnaire provides pertinent data regarding dyspnea, cough, sputum production, sleep disturbances, and physical activity capability, among other aspects.

Presently, COPD is recognized as a systemic disease due to the widespread inflammation it causes [5]. This phenomenon renders the concurrence of other ailments, termed comorbidities, commonplace in these patients [6]. Diabetes mellitus (DM) ranks among the most prevalent comorbidities associated with COPD [7,8].

Obesity, which typically is an unfavorable prognostic factor of various diseases, assumes a paradoxical role in COPD, acting as a favorable prognostic factor [9,10,11]. The mechanism of this phenomenon, coined the obesity paradox, has not been fully elucidated; nonetheless, one prevailing hypothesis suggests that it is, in fact, malnutrition that portends a worse prognosis. Malnutrition is linked to a higher frequency of exacerbations, diminished exercise capacity, heightened mortality rates, and poorer quality of life [12].

However, obesity can contribute to the development of insulin resistance and DM, both prevalent comorbidities in COPD patients [13]. The presence of DM and certain nutritional alterations can cause a deterioration in overall health status and quality of life in patients during both periods of stability and severe exacerbations [14].

The objective of this study was to analyze the influence of DM and nutritional parameters on the exacerbation-associated symptom deterioration and quality of life in COPD patients.

## 2. Patients and Methods

### 2.1. Design

We conducted an observational and noninterventional study involving data collection during patient hospitalization and retrospective review of patients’ clinical characteristics.

### 2.2. Study Population

The patient recruitment period spanned from September 2021 to March 2023. Patients who were admitted with a diagnosis of COPD exacerbation, had a previous diagnosis of COPD using post-bronchodilator spirometry with an FEV1/forced vital capacity (FVC) ratio less than 0.70, and were older than 40 years of age at the time of inclusion were included. Patients with cognitive impairment or other conditions that might hinder completion of the CAT questionnaire, with a previous hospitalization for COPD exacerbation within 2 months prior to the current admission, or with a history of admission to the intensive care unit (ICU) were excluded.

Due to the substantial healthcare burden during the COVID-19 pandemic, not all eligible patients were offered the opportunity to participate in the study.

### 2.3. Data Collection

Upon patient inclusion in the study, anthropometric data (age, gender, weight, height, body mass index (BMI), body fat, muscular mass, and visceral fat), medical histories, and clinical information pertinent to the initial moments of the current admission were gathered.

To measure pulmonary function, bedside spirometry was performed at the time of inclusion and the most recent available spirometry test prior to admission was reviewed.

The CAT questionnaire was administered twice on the day of study enrolment: first, regarding the first day of the current admission and, second, regarding a typical day 2 months before admission.

To assess nutritional parameters, measurements of weight, muscular mass, body fat, and visceral fat using bioimpedance were obtained at the time of inclusion. Additionally, the most recent available blood test results were reviewed to obtain data on hemoglobin, iron, ferritin, and thyroid-stimulating hormone (TSH) levels.

At the time of discharge, the duration of hospitalization was documented.

### 2.4. Statistical Analysis

A descriptive analysis was conducted for all variables included in the study. Qualitative variables were expressed as frequencies and percentages. Quantitative variables with a normal distribution were presented as means and standard deviations (SDs), while those without a normal distribution were expressed as medians and interquartile ranges (IQRs). The normality of distributions was assessed using histograms.

The Friedrich test was used to compare the scores for each item in the CAT questionnaire during periods of stability (baseline CAT) and exacerbations (exacerbation CAT), as well as the total score at the two time points.

Comparison of clinical characteristics, functional parameters, and nutritional data based on whether patients had DM was analyzed using a Student’s *t*-test for quantitative variables or Fisher’s exact test for qualitative variables.

Comparison of CAT questionnaire scores, functional parameters, and hospital stays in relation to each nutritional parameter was conducted using the Mann–Whitney U test.

Multivariate analysis was also conducted to identify and eliminate confounding factors.

For the analysis, multiple discriminant analysis and logistic regression in the case of a single dependent variable were used. And canonical correlation and conjoint analysis were used for multiple dependent variables.

Finally, for quantitative variables, multiple regression and survival analysis were used in the case of a single dependent variable and multivariate analysis of variance (MANOVA) was used for multiple dependent variables.

A two-tailed significance level of *p* < 0.05 was set for all comparisons. All statistical analyses were conducted using IBM SPSS Statistics for Mac, version 26 (IBM Corp., Armonk, NY, USA).

### 2.5. Ethical Considerations

The study received approval from the Ethic Committee for Drug Research at the Gregorio Marañón General University Hospital, with code 14/2019. Written informed consent was obtained from all participants prior to any study-related procedures.

## 3. Results

A total of 50 patients were enrolled in the study. Among them, 30 patients (60%) were male. The mean age was 70.5 years (SD 9.6). At the time of inclusion, 8 patients (16%) were active smokers and 13 patients (26%) had DM. The mean hospital stay was 8.6 days (SD 4.8).

We measured baseline pulmonary function. The mean absolute FEV1 was 1.17 L (SD 0.46) and, expressed as a percentage of the predicted value (FEV1pp), it was 46.7% (SD 0.8). The mean absolute FVC was 2.47 L (SD 0.84) and, expressed as a percentage of the predicted value (FVCpp), it was 75.7% (SD 18.4). The mean absolute FEV1 obtained during an exacerbation was 0.87 L (SD 0.37), and the mean absolute difference compared to baseline FEV1 was −0.36 L (SD 0.34).

Additionally, we measured nutritional parameters. The mean weight was 72.2 kg (SD 15.9), the mean height was 163.6 cm (SD 7.8), and the mean body mass index (BMI) was 27.0 kg/m^2^ (SD 5.6). Based on these data, 30 patients (60%) were overweight and 14 patients (28%) were obese. The mean muscle mass was 29.4% (SD 6.2), the mean body fat was 32.0% (SD 5.6), and the median visceral fat index was 9 (IQR 7–14.5). Accordingly, 28 patients (56%) had low muscle mass, 28 patients (56%) had high body fat, and 16 patients (32%) had a high visceral fat index. The mean hemoglobin level was 13.5 g/dL (SD 1.8), the mean iron level was 91.3 mcg/dL (SD 42.3), the mean ferritin level was 295.3 ng/mL (SD 664.6), and the mean TSH level was 1.57 mUI/L (SD 1.54). According to these data, 13 patients (26%) had anemia, 4 patients (8%) had iron deficiencies, 13 patients (26%) had high ferritin levels, and 5 patients (10%) had high TSH levels.

The median total baseline CAT score was 13.5 (IQR 7–19), and that during an exacerbation was 25 (IQR 17.5–30). The median difference in exacerbation CAT score relative to baseline was 9 (IQR 5–15.25). The scores for each item of the CAT questionnaire, along with the item-wise differences and statistical significance, are presented in Table 1.

Table 2 presents the comparison of clinical, functional, and nutritional parameters based on whether patients had DM. Patients with DM had higher weight (80.1 kg vs. 69.4 kg, *p* = 0.035) and BMI (30.0 kg/m^2^ vs. 25.9 kg/m^2^, *p* = 0.021) than patients without DM. Additionally, diabetic patients had a significantly higher baseline FEV1 than nondiabetic patients (1.38 L vs. 1.09 L, *p* = 0.046).

The comparisons of CAT score differences, FEV1 differences, and hospital stays in patients with and without DM and of varying nutritional parameters are presented in Table 3, Table 4, Table 5 and Table 6. Patients with iron deficiencies, compared to those without, exhibited a significant exacerbation-associated decrease in the CAT score compared to baseline for questions related to daily activity (4.5 vs. 1, *p* = 0.009), feeling secure going out (3 vs. 1, *p* = 0.092), and having energy (3.5 vs. 1, *p* = 0.007), as well as a decrease in the total score (16.5 vs. 9, *p* = 0.041).

The absolute decrease in FEV1 compared to baseline during admission and durations of hospital stays in patients with and without DM and of varying nutritional parameters are shown in Table 7. Patients with DM exhibited a greater decrease in FEV1 during an exacerbation relative to baseline than patients without DM (−0.38 L vs. −0.34 L, *p* = 0.016). Additionally, patients with high TSH levels had shorter hospital stays than those with normal TSH levels (5.0 days vs. 9.0 days, *p* = 0.016).

## 4. Discussion

In our study, exacerbation of COPD was associated with a decrease in CAT questionnaire scores compared to baseline, both in terms of the total score and the individual items included in the questionnaire. Patients differed in clinical characteristics; diabetic patients had higher weight, BMI, and baseline FEV1. Patients with iron deficiencies exhibited greater exacerbation-associated decreases in different CAT questionnaire items and the total score. Additionally, a more significant decrease in FEV1 during a COPD exacerbation was observed in patients with DM, and patients with high TSH levels had shorter hospital stays.

Few published studies have specifically examined exacerbation-associated decreases in CAT questionnaire scores compared to scores during stable periods. However, it has been established that CAT scores are higher in patients with frequent exacerbations than in those without, with a difference of approximately 7 points [15].

In our study, we observed that diabetic patients had higher weight and BMI. This finding is consistent with other published studies [16,17]. This association can be attributed to the link between obesity and insulin resistance [18].

Furthermore, we found that patients with DM had higher baseline FEV1 and a greater decline in FEV1 during a COPD exacerbation than patients without DM. In recent years, several studies have explored the relationship between lung function and DM. A cross-sectional study reported that patients with DM had lower peak flow than those without DM, with no statistically significant differences in FEV1 or FVC [19]. While this finding differs from that of our study, that study did not focus on patients with COPD. The same study showed that the decline in FEV1 was greater in patients with acute glycemic control issues than in those with chronic control issues [19]. In future studies, it would be interesting to associate the degree of glycemic control with the decline in FEV1 during a COPD exacerbation.

In addition, a systematic review indicated that microvascular complications and poor glycemic control were associated with reduced lung function, although it did not conclusively demonstrate that DM itself worsens the decline in lung function [20]. Another longitudinal study demonstrated that DM patients admitted for COPD exacerbations had longer hospital stays [21]. In our study, we did not find a statistically significant difference, likely due to limited statistical power.

However, we observed that patients with iron deficiencies experienced a more significant exacerbation-associated decrease in various CAT questionnaire items, as well as in the total score. The most common cause of anemia worldwide is iron deficiency, and, consequent to anemia, patients may experience increased dyspnea and reduced exercise tolerance [22,23,24].

Finally, a shorter hospital stay was observed in patients with high TSH levels. This finding might be attributed to the relatively advanced age of the patients sampled, as elevated TSH levels in older patients have a cardiovascular protective effect by reducing the risk of arrhythmias [25].

Understanding the nutritional characteristics of our population holds significant implications for enhancing lung health and overall quality of life. By identifying specific nutritional alterations and deficiencies prevalent in the community, tailored interventions can be designed to address these nutritional gaps, thus fostering a healthier and more resilient populace.

Our study has certain limitations. First, the sample size was small, leading to insufficient statistical power when conducting subgroup analyses. This limitation arose from the high healthcare burden during the study period due to the COVID-19 pandemic. Second, because this was a single-center study, the sample might not fully represent the characteristics of the general population of COPD patients. Third, administering the CAT questionnaire retrospectively with reference to the period 2 months before exacerbation could have introduced recall bias. Finally, all diabetic patients had type 2 diabetes. Due to the small sample size, no patients with type 1 diabetes were included, thus precluding the assessment of effects on type 1 diabetic patients.

## 5. Conclusions

In our study, we observed that diabetic patients admitted for a COPD exacerbation experienced a greater decline in FEV1 than those without diabetes, although we did not find longer hospital stays in diabetic patients. Patients with iron deficiencies exhibited a more significant exacerbation-associated decrease in CAT questionnaire scores. In future studies, enhancing statistical power would be important. In clinical practice, for COPD patients admitted due to an exacerbation, metabolic assessment is useful and relevant in the clinical set-up, as endocrinological comorbidities negatively affect clinical and functional aspects of these patients.

## Figures and Tables

**Table 1 jcm-12-06874-t001:** Comparison between baseline CAT questionnaire scores and exacerbation CAT questionnaire scores.

Item	Basal CAT, Median (IQR)	Exacerbation CAT, Median (IQR)	Difference, Median (IQR)	*p*-Value
Cough	1 (0–3)	3 (2–4)	1 (0–2)	<0.001
Sputum	1.5 (0–3)	3.5 (1.75–4.25)	1 (0–2)	<0.001
Tightness	0 (0–1.25)	2 (0–4)	0 (0–2)	<0.001
Dyspnea	3 (2–5)	5 (4–5)	1 (0–2)	<0.001
Activities	1.5 (0–3)	4 (1–5)	2 (0–3)	<0.001
Confidence	0.5 (0–2.25)	3.5 (0.75–5)	1.5 (0–3)	<0.001
Sleep	0 (0–1.25)	1 (0–4)	0 (0–2)	<0.001
Energy	2 (0–3)	3.5 (2.75–5)	2 (0–2)	<0.001
Total Score	13.5 (7–19)	25 (17.5–30)	9 (5–15.25)	<0.001

CAT: COPD assessment test; IQR: interquartile range.

**Table 2 jcm-12-06874-t002:** Comparison of clinical, functional, and nutritional parameters based on the presence of diabetes mellitus.

Variable	DM	Non-DM	*p*-Value
Male, n (%)	9 (69.2)	21 (56.8)	0.522
Age, años (SD)	72.5 (7.1)	69.8 (10.3)	0.401
Weight, kg (SD)	80.1 (13.4)	69.4 (15.9)	0.035
Height, cm (SD)	163.5 (7.5)	163.7 (8.1)	0.960
BMI, kg/m^2^ (SD)	30.0 (4.9)	25.9 (5.5)	0.021
Muscle mass, % (SD)	30.7 (5.5)	28.9 (6.5)	0.382
Body fat, % (SD)	31.8 (10.2)	32.0 (12.8)	0.951
Visceral fat, mean (SD)	13.5 (6.5)	10.1 (5.4)	0.076
Baseline FEV1, L (SD)	1.38 (0.45)	1.09 (0.44)	0.046

DM: diabetes mellitus; SD: standard deviation.

**Table 3 jcm-12-06874-t003:** Comparison of difference in CAT questionnaire score (first items) based on the presence of diabetes mellitus and nutritional parameter alterations (part 1).

Variable, Median (IQR)	Cough	Sputum	Tightness	Dyspnea
DM yes	0 (0–2)	0 (0–3)	0 (0–3)	1 (0–2)
DM no	1 (0–2)	1 (0–2)	0 (0–2)	2 (0–2.5)
*p*-Value	0.081	0.435	0.828	0.404
Overweight yes	1 (0–1)	4 (3–4)	3 (2–3)	2.5 (2–2.5)
Overweight no	0 (0–0)	2 (2–2)	0 (0–0)	2 (2–2)
*p*-Value	0.429	0.472	0.762	0.846
Obesity yes	0.50 (0–2)	1 (0–2.25)	0.5 (0–2.25)	0.5 (0–2.25)
Obesity no	1 (0–2)	1 (0–2)	0 (0–2)	1 (0–2)
*p*-Value	0.267	0.991	0.795	0.858
Low muscle mass	1 (0–2)	1.5 (0–2)	0 (0–2)	1 (0–2)
Normal muscle mass	1 (0–2)	1 (0–2)	0 (0–2)	1 (0–2)
*p*-Value	0.463	0.397	0.877	0.347
High body fat	1 (0–2)	1 (0–2.75)	0.5 (0–2)	1 (0–2)
Normal body fat	1 (0–2)	1 (0.5–2)	0 (0–2)	1 (0–2)
*p*-Value	0.737	0.650	0.594	0.631

IQR: interquartile range; DM: diabetes mellitus.

**Table 4 jcm-12-06874-t004:** Comparison of difference in CAT questionnaire score (first items) based on the presence of diabetes mellitus and nutritional parameter alterations (part 2).

Variable, Median (IQR)	Cough	Sputum	Tightness	Dyspnea
High visceral fat	1 (0–2)	1 (0.25–2)	0 (0–2)	1.5 (0–2)
Normal visceral fat	1 (0–2)	1 (0–2.25)	0.5 (0–3)	1 (0–2)
*p*-Value	0.497	0.958	0.394	0.613
Anemia yes	1 (0–2)	1 (0–2)	0 (0–1)	1 (0–2)
Anemia no	1 (0–2)	1 (0–2.75)	0.5 (0–3)	1 (0–2)
*p*-Value	0.953	0.737	0.117	0.566
Iron deficiency yes	1 (0–3.5)	1 (0–4.25)	0 (0–3.75)	2 (0.25–2.75)
Iron deficiency no	1 (0–2)	1 (0–2)	0 (0–2)	1 (0–2)
*p*-Value	0.841	0.841	0.704	0.546
High ferritin	2 (0.5–2.5)	2 (1–3)	1 (0–2)	1 (0.5–2)
Normal ferritin	1 (0–2)	1 (0–2)	0 (0–2.5)	1 (0–2)
*p*-Value	0.124	0.213	0.856	0.591
High TSH	1 (0–1.25)	1 (0.75–2)	1 (0–4)	1.5 (0.75–2)
Normal TSH	1 (0–2)	1 (0–2)	0 (0–2)	1 (0–2)
*p*-Value	0.608	0.530	0.582	0.635

IQR: interquartile range; TSH: thyroid-stimulating hormone.

**Table 5 jcm-12-06874-t005:** Comparison of difference in CAT questionnaire score (last items) based on the presence of diabetes mellitus and nutritional parameter alterations (part 1).

Variable, Median (IQR)	Activities	Confidence	Sleep	Energy	Total Score
DM yes	2 (0–3)	2 (0–3)	0 (0–1)	2 (1–4)	11 (7.5–14.5)
DM no	2 (0–3)	1 (0–3)	0 (0–2)	1 (0–2)	9 (4.5–17.5)
*p*-Value	0.679	0.890	0.521	0.083	0.602
Overweight yes	3 (2–3)	1 (0–1)	0.5 (0–0.5)	2 (2–2)	17 (4.2–17)
Overweight no	2 (2–2)	4 (4–4)	0 (0–0)	2 (2–2)	8 (8–8)
*p*-Value	0.934	0.658	0.467	0.238	0.796
Obesity yes	1 (0–2.25)	1 (0–2.25)	1 (0–2.25)	2 (0.75–2)	10.5 (4.75–14.5)
Obesity no	2 (0–3)	2 (0–3)	0 (0–1)	1.5 (0–2)	9 (5.25–17.25)
*p*-Value	0.567	0.598	0.110	0.527	0.786
Low muscle mass	2 (0–3)	1 (0–2)	0.5 (0–2)	1 (0–2)	9.25 (5.5–14.75)
Normal mascle mass	2 (0–2.5)	2 (0–3.5)	0 (0–1)	2 (0.5–3)	9 (4.5–15.5)
*p*-Value	0.975	0.585	0.285	0.084	0.685
High body fat	2 (0–2)	1 (0–2)	0 (0–1.75)	1 (0–2)	9 (6.25–13)
Normal body fat	2 (0–3.5)	2 (0–3.5)	0 (0–2)	2 (0–2)	9 (3.5–20)
*p*-Value	0.549	0.468	0.991	0.899	0.879

IQR: interquartile range; DM: diabetes mellitus.

**Table 6 jcm-12-06874-t006:** Comparison of difference in CAT questionnaire score (last items) based on the presence of diabetes mellitus and nutritional parameter alterations (part 2).

Variable, Median (IQR)	Activities	Confidence	Sleep	Energy	Total Score
High visceral fat	0 (0–2.75)	1.5 (0–3)	0 (0–1)	0.5 (0–2)	8.5 (4.25–13.75)
Normal visceral fat	2 (0–3)	1.5 (0–2)	0.5 (0–2)	2 (0.75–2)	9 (5.75–16.5)
*p*-Value	0.304	0.496	0.466	0.099	0.504
Anemia yes	2 (0–2.5)	0 (0–2)	0 (0–2)	1 (0–2)	9 (3.5–9)
Anemia no	2 (0–3)	2 (0–3)	0 (0–2)	2 (0–2)	10.5 (5.5–16.75)
*p*-Value	0.620	0.184	0.961	0.471	0.329
Iron deficiency yes	4.5 (2.5–5)	3 (1.25–4.75)	0.5 (0–2.5)	3.5 (2.25–4.75)	16.5 (11.25–24.75)
Iron deficiency no	1 (0–2)	1 (0–2)	0 (0–2)	1 (0–2)	9 (4–13)
*p*-Value	0.009	0.092	0.947	0.007	0.041
High ferritin	0 (0–3.5)	0 (0–2)	0 (0–3)	1 (0–2)	9 (5–17.5)
Normal ferritin	2 (0–2.5)	2 (0–3)	0 (0–1.5)	2 (0–2)	9 (5–14.5)
*p*-Value	0.535	0.139	0.569	0.159	0.903
High TSH	1 (0–3.25)	1 (0–3.5)	1 (0–2.25)	2 (0.75–2)	9 (2.75–18.5)
Normal TSH	2 (0–3)	1 (0–3)	0 (0–2)	2 (0–2)	10 (4.5–15)
*p*-Value	0.776	0.924	0.776	0.985	0.776

IQR: interquartile range; TSH: thyroid-stimulating hormone.

**Table 7 jcm-12-06874-t007:** Comparison of the absolute FEV1 decline during admission relative to baseline and hospital stay based on the presence of diabetes mellitus and alteration in nutritional parameters.

Variable	FEV1 Difference, L (SD)	Hospital Stay, Days (SD)
DM yes	−0.38 (0.28)	10.3 (4.6)
DM no	−0.34 (0.36)	8.0 (4.8)
*p*-Value	0.016	0.093
Overweight yes	−0.24 (0.27)	10.0 (2.8)
Overweight no	−0.38 (0.23)	16.0 (1.4)
*p*-Value	0.432	0.777
Obesity yes	−0.45 (0.41)	7.7 (3.3)
Obesity no	−0.32 (0.30)	9.0 (5.3)
*p*-Value	0.610	0.632
Low muscle mass	−0.37 (0.35)	8.3 (5.0)
Normal muscle mass	−0.29 (0.22)	9.2 (4.6)
*p*-Value	0.235	0.463
High body fat	−0.36 (0.32)	8.3 (4.5)
Normal body fat	−0.29 (0.28)	9.2 (5.2)
*p*-Value	0.547	0.584
High visceral fat	−0.35 (0.37)	8.9 (4.6)
Normal visceral fat	−0.36 (0.32)	8.5 (5.2)
*p*-Value	0.415	0.835
Anemia yes	−0.32 (0.25)	9.6 (5.8)
Anemia no	−0.37 (0.37)	8.3 (4.4)
*p*-Value	0.711	0.519
Iron deficiency yes	−0.28 (0.18)	12.3 (6.5)
Iron deficiency no	−3.39 (0.37)	8.4 (4.4)
*p*-Value	0.858	0.199
High ferritin	−0.44 (0.37)	7.7 (4.3)
Normal ferritin	−0.33 (0.32)	8.9 (4.9)
*p*-Value	0.634	0.568
High TSH	−0.54 (0.53)	5.5 (1.8)
Normal TSH	−0.36 (0.34)	10.0 (4.8)
*p*-Value	0.288	0.016

FEV1: forced expiratory volume in first second; SD: standard deviation; DM: diabetes mellitus; TSH: thyroid-stimulating hormone.

## Data Availability

The data that support the findings of this study are available from the corresponding author upon reasonable request.

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
