# Peer review of "Influence of Diabetes Mellitus and Nutritional Parameters on Clinical and Functional Aspects and Quality of Life in Patients Hospitalized Due to Exacerbation of Chronic Obstructive Pulmonary Disease"

_jcm, 2023, doi:10.3390/jcm12216874_

Round 1

Reviewer 1 Report

Comments and Suggestions for Authors

Article

Influence of diabetes mellitus and nutritional parameters on

Clinical and functional aspects and quality of life of patients

Hospitalized for exacerbation of chronic obstructive pulmonary disease

  The observational study dealt with patients with chronic obstructive pulmonary disease (COPD). During severe exacerbations, nutritional and endocrinological comorbidities can play an important role in the clinical and functional aspects of these patients. The aim of this study was to analyze the influence of the presence of diabetes mellitus (DM) and nutritional parameters. on worsening symptoms and quality of life during a severe exacerbation in patients with COPD. An observational study was conducted on patients with COPD who were admitted for an exacerbation. The COPD Assessment Questionnaire (CAT) was applied and clinical and functional parameters were compared based on the presence of nutritional and endocrinological changes. For patients with COPD who are admitted due to an exacerbation, it is crucial to consider endocrinological comorbidities, as they negatively affect the clinical and functional aspects of these patients.

Correctly set methodologically, precisely defined goals, well done statistically, with very useful results. the discussion followed similar studies in the world, and the references are adequate for the written text general impression - well written clinical article

I leave the decision on acceptance to the editor, my opinion is already - very well done

Author Response

We appreciate the Reviewer's comment and we completely agree with the Reviewer regarding the importance of the nutritional aspect of COPD patients who suffer a severe exacerbation.

Reviewer 2 Report

Comments and Suggestions for Authors

I have the following comments:

1. Re-write the first sentence in the abstract and introduction; replace "characterized by"

2. Cite appropriate reference for the diagnosis of COPD, like GOLD 2023.

3. The methods section needs more details.

4. Table 1: transfer "median (IQR)" under CAT column

5. Table 2: Baseline FEV1 was significantly higher in DM group than non-DM, what is the interpretation? what is mentioned in the discussion is not enough

6. The sample size is very small

7. It is hard to follow and understand Tables 3, 4 and 5, and there is problems in the design. Add some details in the legend. Although you mentioned the statistical analysis details in the methods, it is better to mention the sraristical test used in each part.

8. The cited references are not enough.

9. I can not find a real novelty in this study.

10. You need to write in discussion what are the implications of this study?

Comments on the Quality of English Language

Needs some improvement

Author Response

Comment 1: Re-write the first sentence in the abstract and introduction; replace "characterized by"

Response 1: Taking into account the Reviewer’s comment, we rewrote the first sentence in the Abstract and Introduction section.

Patients with chronic obstructive pulmonary disease (COPD) may experience exacerbations.

Comment 2: Cite appropriate reference for the diagnosis of COPD, like GOLD 2023.

Response 2: GOLD 2023 was previously cited, but we added a paragraph in the Introduction section to describe the diagnosis of COPD.

Chronic obstructive pulmonary disease (COPD) is characterized by a chronic and not fully reversible limitation of airflow due to alterations in the airways, which is accompanied by persistent respiratory symptoms caused by significant exposure to harmful gases or particles [1].

Comment 3: The methods section needs more details.

Response 3: Taking into account the Reviewer’s comment, we included more details in the Methods section.

Comment 4: Table 1: transfer "median (IQR)" under CAT column

Response 4: The change was made according to the Reviewer’s comment.

Comment 5: Table 2: Baseline FEV1 was significantly higher in DM group than non-DM, what is the interpretation? what is mentioned in the discussion is not enough

Response 5: We appreciate the reviewer's comment. We performed a bibliographic review, and no explanation for this finding was found. We think that coincidentally there has been a difference between both groups.

Comment 6: The sample size is very small

Response 6: We appreciate the Reviewer’s comment. The small sample lead to an insufficient statistical power when conducting subgroup analyses. That is the reason because we mentioned it in the Discussion section.

Comment 7: It is hard to follow and understand Tables 3, 4 and 5, and there is problems in the design. Add some details in the legend. Although you mentioned the statistical analysis details in the methods, it is better to mention the statistical test used in each part.

Response 7: We understand the Reviewer’s concern. The tables are extensive, and we have problems to edit in the word page. We performed some changes to the format with our capacities, and we hope that the Tables are perfect in the final edition.

Comment 8: The cited references are not enough.

Response 8: Taking into account the Reviewer’s comment, we added more cited references.

3. Vogelmeier CF, Román-Rodríguez M, Singh D, Han MK, Rodríguez-Roisin R, Ferguson GT. Goals of COPD treatment: Focus on symptoms and exacerbations. Respir Med. 2020, 166, 105938.

8. Natali D, Cloatre G, Hovette P, Cochrane B. Screening for comorbidities in COPD. Breathe (Sheff). 2020, 16, 190315.

17. Aras M, Tchang BG, Pape J. Obesity and Diabetes. Nurs Clin North Am. 2021, 56, 527-541.

24. Baltierra D, Harper T, Jones MP, Nau KC. Hematologic Disorders: Anemia. FP Essent. 2015, 433, 11-15.

Comment 9: I cannot find a real novelty in this study.

Response 9: This study introduces a novel approach by examining the nutritional aspects within our population. It aims to identify variables in routine clinical practice that can be addressed to enhance respiratory health and overall quality of life.

Comment 10: You need to write in discussion what are the implications of this study?

Response 10: Taking into account the Reviewer’s comment, we added a new paragraph in the Discussion section.

Understanding the nutritional characteristics of our population holds significant implications for enhancing lung health and overall quality of life. By identifying specifics nutritional alterations and deficiencies prevalent in the community, tailored interventions can be designed to address these nutritional gaps, thus fostering a healthier and more resilient populace.

Round 2

Reviewer 2 Report

Comments and Suggestions for Authors

The comments regarding the tables were not fully addressed

Author Response

Comment 1: The comments regarding the tables were not fully addressed

Response 1: We have divided the tables for easier reading.
